# Human Umbilical Cord Mesenchymal Stem Cells Protect against Renal Ischemia-Reperfusion Injury by Secreting Extracellular Vesicles Loaded with miR-148b-3p That Target Pyruvate Dehydrogenase Kinase 4 to Inhibit Endoplasmic Reticulum Stress at the Reperfusion Stages

**DOI:** 10.3390/ijms24108899

**Published:** 2023-05-17

**Authors:** Wei Shi, Xiang Zhou, Xinyuan Li, Xiang Peng, Guo Chen, Yang Li, Chunlin Zhang, Haitao Yu, Zhenwei Feng, Xin Gou, Jing Fan

**Affiliations:** 1Department of Urology, The First Affiliated Hospital of Chongqing Medical University, Chongqing 400016, China; swurology@outlook.com (W.S.);; 2Chongqing Key Laboratory of Molecular Oncology and Epigenetics, The First Affiliated Hospital of Chongqing Medical University, Chongqing 400016, China

**Keywords:** human umbilical cord mesenchymal stem cells, apoptosis, extracellular vesicles, ischemia/reperfusion injury, endoplasmic reticulum stress, MicroRNA

## Abstract

Renal ischemia-reperfusion (I/R) injury is a leading cause of acute kidney injury (AKI), with high mortality. Recent studies have reported that human umbilical cord mesenchymal stem cells (HucMSCs) play an important role in repairing organ and tissue injuries because of their unique characteristics. However, the potential of HucMSC extracellular vesicles (HucMSC-EVs) to promote the repair of renal tubular cells remains to be explored. This study found that HucMSC-EVs derived from HucMSCs played a protective role and were associated with kidney I/R injury. We found that miR-148b-3p in HucMSC-EVs had a protective effect against kidney I/R injury. HK-2 cells overexpressing miR-148b-3p were protected against I/R injury by inhibiting apoptosis. Next, the target mRNA of miR-148b-3p was predicted online, and the target mRNA, pyruvate dehydrogenase kinase 4 (*PDK4*), was identified and verified using dual luciferase. We discovered that I/R injury significantly increased endoplasmic reticulum (ER) stress, whereas siR-PDK4 inhibited these effects and protected against I/R injury. Interestingly, after administrating HucMSC-EVs to HK-2 cells, *PDK4* expression and ER stress induced by I/R injury were significantly inhibited. HK-2 ingested miR-148b-3p from HucMSC-EVs, and its ER induced by I/R injury was significantly deregulated. This study suggests that HucMSC-EVs protect kidneys from I/R injury during the early I/R stage. These results suggest a new mechanism for HucMSC-EVs in treating AKI and provide a new treatment strategy for I/R injury.

## 1. Introduction

Acute kidney injury (AKI), characterized by high mortality rates, is on the rise in numerous countries [1]. Previous studies have revealed that AKI is histologically characterized by apoptosis and inflammation in tubular cells [2]. AKI is usually caused by kidney ischemia/reperfusion (I/R), kidney toxins, and sepsis, and kidney I/R is the most common cause of AKI [3]. Kidney I/R injury after transplantation has been identified as a serious clinical problem, as it triggers an acute inflammatory response followed by rapid loss of kidney function, affecting the prognosis of the transplanted kidney.

Human umbilical cord mesenchymal stem cells (HucMSCs) have been extensively investigated for their self-renewal and pluripotency, ability to differentiate into osteoblasts, chondrocytes, and adipocytes, as well as ability to secrete cytokines [4]. Exogenous HucMSCs improved AKI and glomerular filtration function [5]. Interestingly, HucMSCs inhibit astrocyte activation, promote repair of associated hypoxic/ischemic brain injury, and protect the hippocampus from brain I/R injury in rats [6]. Thus, HucMSC therapy had a significant protective effect against I/R injury. It is now widely believed that MSC ameliorates disease by secreting paracrine and stimulating host cells rather than directly implanting and replacing them. In this context, there is increasing evidence that stems cell-derived extracellular vehicles (EVs) mediate the carrying and transfer of shuttle, such as regulatory miRNA, cytokines, growth factors, and signaling lipids [7].

All prokaryotes, eukaryotes, and other cells release membrane-containing vesicles in an evolutionarily conserved way, and EVs are secreted by cells into the extracellular space [8,9,10]. EVs consist of heterogeneous membrane vesicles from different sources, including exosomes and microvesicles. They are of different sizes, usually between 50 and 500 nm. EVs can carry various cargo, including proteins, lipids, and nucleic acids, which are involved in intercellular communication, as well as in normal cellular homeostasis or pathological development as signal carriers, and these contents vary according to different cells and conditions [11]. Recently, evidence has displayed that exosomes secreted by HucMSCs can reduce I/R injury and promote the regeneration and repair of renal tubules [12]. MicroRNAs (miRNAs) are small, ~21–22-nt non-coding RNAs that regulate target gene expression by identifying homologous sequences and interfering with transcriptional, translational, or epigenetic processes [13], which constitute an important part of EV. Several studies have revealed the expression of HucMSCs in kidney I/R injury. However, the mechanisms of targeting different miRNAs are diverse and complex. Therefore, further studies on the mechanisms of human umbilical cord mesenchymal stem cells extracellular vesicles (HucMSC-EVs) are urgently required.

Many genetic and environmental impairments impede the ability of cells to properly fold and post-translationally fold secretory and transmembrane proteins in the endoplasmic reticulum, resulting in the accumulation of misfolded proteins in this organelle, a condition called endoplasmic reticulum (ER)stress [14]. ER stress, also known as unfolded protein response (UPR), is triggered by three parallel ER transmembrane protein-activated signaling pathways that mediate UPR: IRE1, PERK, and ATF6, to restore ER homeostasis and ensure intracellular balance [15]. Several studies have revealed that ER stress is associated with many kidney diseases, including primary glomerulonephritis, acute kidney injury, diabetic nephropathy, and renal fibrosis [16]. Therefore, whether ER stress is involved in the effects of HucMSCs on renal I/R injury requires further investigation. In addition, ER stress is activated at the beginning of the pathological process. ER stress is significantly induced 4–6 h after renal I/R [17].

Pyruvate dehydrogenase kinase 4 (*PDK4*) is a mitochondrial matrix enzyme that regulates the activity of pyruvate dehydrogenase complex. It is also a key regulator of pyruvate oxidation and glucose homeostasis [18]. Previous studies have found that *PDK4* inhibition can increase myocardial glucose uptake after myocardial I/R, which can effectively promote the cardiac function, while for acute renal injury, cisplatin-induced renal tubular necrosis is attributed to the increase of *PDK4* expression and activity [18,19]. However, the role of *PDK4* in endoplasmic reticulum stress and the molecular mechanism of its role in ischemia-induced renal injury are still unclear. We discussed the role of *PDK4* in ER stress and the targeting relationship between miR-148b-3p and *PDK4*.

This study aimed to investigate the protective effects of HucMSC-EVs on kidney I/R injury in vivo and in vitro. In addition, we explored the mechanism of HucMSC-EVs, demonstrating for the first time that HucMSC-EVs can deliver miR-148b-3p to renal tubular epithelial cells and inhibit apoptosis by silencing the expression of *PDK4* and ER stress. These findings identify a novel resistance mechanism to kidney I/R injury and provide a potential therapeutic approach for the clinical treatment of AKI.

## 2. Results

### 2.1. Isolation and Identification of HucMSC-EVs

First, we isolated HucMSC-EVs by hyper centrifugation and then identified EVs secreted by HucMSCs. The main shapes of HucMSC-EVs were circular and double-coated by transmission electron microscopy (Figure 1a). NTA (nanoparticle tracking analysis) detected HucMSC-EVs with an average diameter of 147.5 nm (Figure 1b) and a single peak in intensity size distribution, which confirmed the high purity of isolated EVs. HucMSC-EVs were successfully isolated. Protein markers of EVs (CD9 and TSG101) were significantly higher in HucMSC-EVs than in HucMSCs, and the protein negative marker of EVs (CALNEXIN) was significantly higher in HucMSCs than in HucMSC-EVs (Figure 1c). PKH membrane dyes are used in EV studies and their function because of their nonspecific binding to lipid bilayers or protein components and their simple and rapid staining procedures, such as PKH67 and PKH26 [20]. To demonstrate that HucMSC-EVs can be ingested by HK-2 cells, we incubated HucMSC-EV-labeled PKH67 (green) with HK-2 for 24 h, and then observed them using a laser scanning confocal microscope. Many green particles were observed in the cytoplasm of the HK-2 cells. Therefore, it was confirmed that HucMSC-EVs could be absorbed into ActinRed-labeled HK-2 cells and play a role (Figure 1d).

To investigate whether HucMSC-EVs protected the kidneys by inhibiting I/R-induced apoptosis, we used flow cytometry to measure the apoptosis rate of HK-2 cells treated under different conditions. The results depicted a significant increase in apoptosis in cells treated with I/R compared to control cells. However, co-culture with HucMSCs and treatment with HucMSC-EVs significantly inhibited cellular apoptosis (Figure 2a). Meanwhile, we found that HK-2 cells treated with I/R markedly increased the expression levels of cleaved caspase-3 and Bax and decreased the expression level of Bcl-2. Co-culturing with HucMSCs or treatment with HucMSC-EVs reversed the expression of these proteins (Figure 2b). In addition, we found that EV-removed CM group was significantly less potent against apoptosis than HucMSCs and HucMSC-EVs, whereas there were no obvious differences between HucMSC and HucMSC-EV groups in Bax and cleaved caspase-3 levels, indicating that the protective effects of HucMSCs were mediated by their EVs. These results suggested that HucMSC-EVs play a key protective role against I/R injury induced by HK-2 cell apoptosis.

### 2.2. miR-148b-3p Participates in the Protective Effect of HucMSC-EVs on Renal Ischemia

A recent study identified the miRNA landscape in HucMSC-EVs and showed that miR-148b-3p is enriched [21]. Thus, we measured miR-148b-3p expression levels in HucMSC-EVs and HK-2 cells, the results showed that the expression level of miR-148b-3p in HucMSC-EVs were higher than those in HK-2 cells (Figure 3a). Furthermore, we detected miR-148b-3p expression levels in HK-2 cells after I/R treatment at different time points and demonstrated that miR-148b-3p levels were significantly reduced after I/R treatment (Figure 3b). These data demonstrated that HK-2 cells contain miR-148b-3p. Therefore, we investigated the role of miR-148b-3p in I/R injury. First, miR-148b-3p mimics and inhibitors were administered to HK-2 cells, followed by I/R treatment. The results demonstrated that mimics and inhibitors of miR-148b-3p upregulated and downregulated their expression in HK-2 cells (Figure 3c). miR-148b-3p overexpression in HK-2 cells inhibits I/R injury-induced apoptosis. The expression levels of cleaved caspase-3 and Bax were decreased, whereas that Bcl-2 was increased. However, miR-148b-3p knockdown in HK-2 cells after I/R injury reversed the expression of these apoptosis-related proteins (Figure 3d). Accordingly, miR-148b-3p in EVs play a critical role in the HucMSC-EV-mediated inhibition of I/R-induced apoptosis.

### 2.3. miR-148b-3p Regulates PDK4 and Participates in the Protective of I/R Injury

In order to further probe the molecular mechanisms of miR-148b-3p function in cell apoptosis, we screened for miR-148b-3p target genes on several websites, including PITA, PicTar, TargetScan, and miRanda (Figure 4a). In total, we identified 10 target genes predicted by the four online databases, of which DEDD, KIT, PIK3R3, and *PDK4* were found to related to the apoptosis of I/R from literature review. Next, we used mimics to transfect miR-148b-3p into HK-2 cells and detected whether it could regulate in the expression of these mRNAs. The level of *PDK4* was the most significantly changed (Figure 4b). We predicted putative binding sites of miR-148b-3p and *PDK4* using starBase database. Notably, the luciferase reporter assay demonstrated that luciferase reporter activity was significantly reduced by miR-148b-3p overexpression compared to miRNA-NC. Furthermore, the activity of the *PDK4*-3′-UTR-mutant luciferase reporter was unaffected by the mimics of miR-148b-3p, suggesting that *PDK4* was a direct target of miR-148b-3p (Figure 4c,d).

To prove that HucMSC-EVs could rescue I/R injury and reduce apoptosis via *PDK4* signaling, we investigated the interaction between miR-148b-3p and *PDK4* in kidney I/R injury. miR-148b-3p inhibited *PDK4* protein expression in I/R injury of HK-2 cells, while miR-148b-3p inhibitor promoted *PDK4* protein expression (Figure 4e). These findings suggest that the direct downregulation of *PDK4* by miR-148b-3p could be a potential protective mechanism.

### 2.4. PDK4 Aggravated the I/R Injury in HK-2 Cells by ATF-6 Pathway of ER Stress

*PDK4* is a key enzyme that regulates pyruvate dehydrogenase complex (PDC), oxidation of pyruvate, and glucose homeostasis [22]. Previous studies have revealed that *PDK4* inhibition can increase myocardial glucose uptake after myocardial I/R, effectively promoting cardiac function recovery [18]. *PDK4* protects against MI/R injury [23]. Therefore, we hypothesized that miR-148b-3p derived from EVs inhibits *PDK4* and downregulates the ATF-6 pathway during ER stress.

First, in order to verify our assumption, we use western blotting to detect the level of *PDK4* expression in HK-2 cells in different groups. Moreover, I/R injury activated ATF-6 pathway of ER stress, but *PDK4*- knockdown reduced the expression of ER stress (Figure 4f).

HK-2 cells co-cultured with non-EVs CM, HucMSCs, and HucMSC-EVs were processed I/R treatment for 24 h. Western blotting was performed to examine the relative protein expression. ATF-6, GRP78/BIP, and CHOP expression levels dramatically increased after I/R treatment, and their protein levels were downregulated (Figure 4g).

These findings proved that miR-148b-3p inhibits the ATF-6 pathway of ER stress and protects against I/R injury by suppressing *PDK4* expression.

### 2.5. HucMSC-EVs Alleviate Kidney I/R Injury In Vivo

In order to evaluate the role of HucMSC-EVs in kidney I/R injury, the kidney I/R model of mice were established. The HucMSC-EVs were injected through the tail vein. In order to verify whether HucMSC-EVs could present in kidney organs, PKH26 (EV labeled) revealed a significant accumulation of HucMSC-EVs that accumulated in the kidney compared to the PBS group (Figure 5a).

As displayed in Figure 5b,c, the levels of serum CREA and BUN, which reflect renal function, in group I/R+PBS were significantly higher than those in group I/R+EVs. Furthermore, HucMSC-EVs improved tubular dilatation, brush border loss, nuclear loss, and cast formation caused by I/R injury (Figure 5d). In addition, the levels of cleaved caspase-3 and *PDK4* in vivo were detected by IHC. In addition to the in vitro experiments, the levels of cleaved caspase-3 and *PDK4* were higher after I/R injury, and the expression of HucMSC-EVs was reduced (Figure 5e,f). These results indicated that HucMSC-EVs can mitigate I/R injury and downregulate the level of *PDK4*.

## 3. Discussion

In this study, we demonstrated that EVs secreted by HucMSCs protected against I/R injury in renal tubule epithelial cells in vitro and in vivo. Most importantly, we explored a novel mechanism whereby HucMSC-EVs transfer miR-148b-3p to renal cells by inhibiting ER stress through downregulating the expression of *PDK4* and ATF-6 pathways. These viewpoints may help us to elucidate the protective mechanism of HucMSC-EVS in kidney disease and provide a theoretical basis for the use of HucMSC-EVS in the treatment of AKI.

HucMSCs are effective in treating I/R injury. In contrast to bone marrow stem cells, HucMSCs have characteristics, such as non-invasive harvesting and rapid self-renewal, and are increasingly used by researchers [5,24]. Recent studies have displayed that MSC paracrine mechanisms have received much attention, especially EVs. EVs as biomolecular carrier mediate intercellular communication. Therefore, studies have shown that the protective effect of MSCs against I/R injury is mediated by EVs [25]. Proteomic studies have demonstrated that HucMSC-EVs are prominent in tissue damage repair [26].

miRNAs have been recognized as key and the most studied molecules in EVs and miRNAs packaged in EVs have regulatory functions in development, differentiation, angiogenesis, proliferation, apoptosis, and many other pathological processes [27,28]. This study demonstrated that EV-derived miR-148b-3p inhibited I/R-induced apoptosis in renal cells. Moreover, previous studies have revealed that miR-148b-3p deficiency in cardiomyocytes increases the risk of apoptosis induced by I/R injury, while the overexpression of miR-148b-3p downregulates the pro-apoptotic gene, SIRT7 [29]. Furthermore, increased miR-148b-3p expression has been observed in kidney I/R injury, suggesting that it may limit renal injury. We also explored the downstream mechanisms of miR-148b-3p and found that it downregulates the *PDK4* expression. *PDK4*, a key enzyme regulating the activity of pyruvate dehydrogenase complex, plays an important role not only in many physiological developmental processes, but also in the pathological processes of different diseases. In renal ischemia, improving blood urea nitrogen/serum creatinine levels by *PDK4* may reduce glomerular injury [19,22]. A previous study displayed the anti-apoptotic effects of *PDK4* [30]. Downregulation of *PDK4* inactivation in AKI may have therapeutic potential in preventing cisplatin-induced kidney injury, providing a novel therapeutic approach for patients with AKI [19]. Our recent study demonstrated that HucMSC-EVs significantly reduce *PDK4* expression caused by I/R injury. Additionally, we report, for the first time, that miR-148b-3p participates in HucMSC-EVs inhibition of *PDK4* expression.

Although miR-148b-3p significantly inhibited the expression of *PDK4* in the miR-148b-3p mimics group, it markedly upregulated *PDK4* expression in the miR-148b-3p knockdown group. Thus, we hypothesized that packaging miR-148b-3p into HucMSC-EVs would be sufficient to inhibit *PDK4* expression.

There is increasing evidence linking ER stress to apoptosis. Previous studies have indicated that ATF6 is regulated by lymphoid enhancer-binding factor 1 and that ATF6 binds to Bcl-2 promoter to further regulate apoptosis [31]. Furthermore, ERK pathway is arelated to apoptosis. Recent studies have demonstrated that miR-211-5p effectively reduces spinal cord injury-induced neuronal apoptosis and inflammation by directly targeting ATF-6 and regulating ER stress [32]. In this study, ATF-6 signaling pathway was activated in HucMSCs and HucMSC-EVs groups, while the expression of cleaved caspase-3 and Bax proteins decreased and that of Bcl-2 protein increased. Interestingly, we found an increase in ATF-6 expression compared to the untreated group in co-cultures with non-EV CM after ischemic reperfusion, which we hypothesized could be due to the effective protection of HK-2 cells on stress, which was amplified after treatment with HucMSC-EVs. Our further study depicted that miR-148b-3p packaged in HucMSC-EVs activates ATF-6 pathway during ER stress by regulating *PDK4*.

Taken together, our study demonstrated for the first time that miR-148b-3p in HucMSC-EVs inhibits apoptosis in kidney I/R injury. Additionally, we explored the potential molecular mechanisms by which miR-148b-3p downregulated *PDK4* expression, activated ATF-6 pathway, and induced ER stress. Such findings suggest a novel possibility that HucMSC-EVs protect against AKI and establish a new link between miR-148b-3p, *PDK4*, and ATF-6 in kidney I/R injury.

## 4. Materials and Methods

### 4.1. Animals, Cells and Reagents

C57BL/6 male mice (20–25 g, 8–10 wk old) were provided by the Animal Experiment Central at Chongqing Medical University (Chongqing, China). The ethical approval number of our institute is:2022-k276. HucMSCs were obtained from the Zhong Qiao Xin Zhou Biotechnology Co., Ltd. (Shanghai, China), and the human renal tubular epithelial cell line HK-2 was purchased from the Procell Life Science&Technology Co., Ltd. (Wuhan, China). antimycin A was purchased from Maokang Biotechnology Co., Ltd. (Shanghai, China), 2-Deoxy-D-glucose was purchased from Sigma Aldrich (St. Louis, MO, USA), Fetal bovine serum (FBS), PKH26, and PKH67 were purchased from Millipore Sigma (Burlington, MA, USA). The pAb anti-cleaved caspase-3 (9664S), Bcl-2 (3498), GRP78 (3177), IRE1-α (3294), XBP-1S (83418), CHOP (2895), p-PERK (3179), ATF6 (65880), and β-actin (3700) were obtained from Cell Signaling Technology (Danvers, MA, USA). The pAb anti-PDK4 (YN5701) was obtained from ImmunoWay Biotechnology Company (Plano, TX, USA). The pAb anti-TSG101 (28283-1-AP) and CD9 (20597-1-AP) were obtained from Proteintech Group, Inc. (Wuhan, China). The mimic miRNAs, inhibitor miRNAs, and small interfering RNAs (siRNAs) were purchased from Tsingke (Beijing, China). Lipofectamine 2000 was purchased from Thermo Fisher Scientific (Waltham, MA, USA).

### 4.2. Cell Culture Experiments

The current cell experiments included these groups: control, ischemia (24 h reperfusion), non-HucMSC-Evs CM, HucMSC-Evs (24 h reperfusion), HucMSC mimics, HucMSC inhibitors, and PDK4 siRNAs.

The HucMSCs and HK-2 cells were cultured in F12 medium at 37 °C and 5% CO_2_ (growth medium), which contained 10% FBS and 1% penicillin-streptomycin. In order to simulate the case of I/R, we set up the oxygen-glucose deprivation (OGD) [33], a chemical anoxia/reoxygenation model by culturing the HK-2 cells in a hypoxic solution containing glucose-free medium and antimycin A (5 mM) and 2 Deoxy-D-glucose (a kind of glycolysis inhibitor, 5 mM) at 37 °C with 5% CO_2_. The cells were incubated with hypoxic solution for 45 min before collection. After 45 min, cells were cultured in freshoxygenated culture medium, which included a variety of EVs, inducers, or inhibitors, to regain the oxygen supply in other groups.

### 4.3. Cell Transfection

The miRNAs were screened by miRmap (https://mirmap.ezlab.org/docs/), accessed on 7 March 2022, TargetScan (http://www.targetscan.org), accessed on 7 March 2022, PicTar (http://www.pictar.org/), accessed on 7 March 2022, PITA (https://genie.weizmann.ac.il/pubs/mir07/mir07_prediction.html), accessed on 8 March 2022, and PubMed (https://www.ncbi.nlm.nih.gov) accessed on 9 March 2022. The sequences of mimics, inhibitors of miRNA and siRNA of PDK4 were acquired from Tsingke (Beijing, China). Moreover, a scrambled siRNA was synthesized as negative control (NC-si). We performed cell transfection with Lipofectamine 2000. The previous culture medium was replaced by serum-free medium, and Lipofectamine 2000 mimics and inhibitor siRNAs were mixed into the serum-free medium, respectively. Afterwards, the transfection solution was replaced by fresh growth medium, and these cells were cultured for another 24 h before the next process.

### 4.4. Animal Experiments

All animal procedures comply with the Animal Management Rule of the Ministry of Health, P.R. China; 55, 2001 and the Guide for the Care and Use of Laboratory Animals (National Institutes of Health, Bethesda, MD, USA). Animals were housed in pathogen-free conditions under a controlled 24 h light-dark cycle and allowed free access to water and chow. The mice (*n* = 5/group) were divided into groups: sham; PBS (groups without the EVs during the reperfusion stage); and HucMSC-EVs (groups with the administration of 5 × 10^10^ particles/100 mL during the reperfusion phase). The renal I/R injury model was established as follows. Mice were anaesthetized by intraperitoneal injection of pentobarbital sodium (75 mg/kg), and the animals were placed on a warmed surgical pad to keep the body temperature at 36.5–37.5 °C during surgery. After exposure of the kidneys on the dorsal side, the renal artery and vein were dissected, clamped for 45 min, and the clamps were removed non-traumatically. The label of successful reperfusion is the immediate change of the kidney color from atropurpureus to red. The mice in the HucMSC-EVs group received EVs injections via the tail vein 1 h before I/R, and the mice in other groups were injected with sterile PBS via tail vein simultaneously. The mice in the non-HucMSC-EVs group and HucMSC-EVs group were euthanized after 24 h of reperfusion. Postoperatively, blood samples were taken to measure blood urea and serum creatinine (CREA). The kidneys were subject to quantitative analysis by harvesting for hematoxylin-eosin (HE) staining and immunohistochemistry (IHC). The serum and kidneys of all mice were collected and stored at 4% paraformaldehyde or at −80 °C.

### 4.5. Quantitative Real-Time Polymerase Chain Reaction

Total RNA was extracted from HK-2, and cells were preprocessed according to the instructions in the RNA Extraction Kit (Takara, Japan). Purified total RNA (1 μg) was reverse-transcribed to cDNA using the PrimeScript RT reagent kit (Takara, Japan). Quantitative real-time PCR (qRT–PCR) was performed using a SYBR (R) Prime-Script RT–PCR kit (Takara, Japan) and an ABI 7500 sequence detection system (Applied Biosystems, CA, USA). Each gene reaction was performed in triplicate, and the gene expression level was determined by the internal control β-actin. Primer sequences are shown in Table 1.

### 4.6. Dual-Luciferase Reporter Assay

PDK4 fragments containing miR-148b-3p binding sites were cloned into pmirGLO oligosaccharide enzyme vectors (Promega, Madison, WI, USA), and then pmirGLO-PDK4-wild type (Wt) reporting vectors were constructed. Based on pmirGLO-PDK4-Wt sequence, a pmirGLO-PDK4 mutant (mut) report vector was constructed using a mutant binding site of miR-148b-3p. The constructed vector was transfected into 293T cells, followed by miR-148-3p and miR-NC, respectively. After 48 h, luciferase activity was measured using a dual-luciferase report analysis system (Promega), and relative activity was expressed as the ratio of luciferase activity in fireflies to Renilla luciferase activity.

### 4.7. Biochemical Analysis

After general anesthesia by the intraperitoneal injection of pentobarbital sodium, the blood samples were collected and serum was collected by centrifugation (7500× *g*, 15 min), The concentrations of urea and CREA in each group were detected by the Biochemical Laboratory of Sichuan Scientist Biotechnology Co., Ltd. (Sichuan, China), and 5 samples were collected from each group.

### 4.8. HE Staining

The kidneys of all mice were fixed in 4% formaldehyde, dehydrated in a graded ethanol series and xylene, and then embedded in paraffin. Slices (4 mm) were soaked in hematoxylin for 5 min and rinsed with cold running water. After incubation with 1% hydrochloric acid alcohol, cells were washed and dyed with 0.5% eosin dye solution. Finally, 5 fields were randomly selected from each section.

### 4.9. Immunohistochemical Staining

Paraffin-embedded renal tissue sections were dewaxed with xylene and rehydrated with low concentration ethanol, and then immersed in sodium citrate buffer with heating in a steam vessel for 30 min for antigen retrieval. The endogenous peroxidase activity was then eliminated with 3% hydrogen peroxide, and 15-min sections were blocked with normal goat serum at room temperature. The sections were incubated with the primary antibodies overnight at 4 °C, and the 30-min sections were incubated with the secondary antibodies at 37 °C. Then, the 30-min sections were incubated with streptavidin-horseradish peroxidase conjugate at 37 °C. The sections were stained with DAB for approximately 1 min and hematoxylin for 10 s. In order to determine of the expression levels of various markers, Image-Pro Plus (IPP, v.6.0.206; Media Cybernetics, Silver Spring, MD, USA) was used for quantitative analysis.

### 4.10. Small Extracellular Vesicles Isolation

The EVs in FBS were removed by 140,000× *g* ultracentrifugation for 18 h. HucMSCs were added to 15-cm culture plates containing 10% FBS, and the cells were grown to 70%–80% fusion. The conventional medium was then removed, and EVs-depleted 10% FBS-containing medium was added. The gold standard for the separation of EVs is differential hypervelocity centrifugation [34]. The HucMSCs culture medium was collected 48 h later, and differential centrifugation was adopted to separate the EVs from the HucMSCs medium. The centrifugation conditions were as follows: 500× *g* for 10 min, 2000× *g* for 10 min, and 12,000× *g* for 30 min at 4 °C, and the supernatant was preserved each time. Then, the supernatant was centrifuged for 120,000× *g* at 4 °C for 70 min, the supernatant was collected as non-extracellular vesicle conditioned medium (non-EVs-CM), and pellets were resuspended in PBS to reduce the remaining solubility factor, before another ultracentrifugation at 120,000× *g* for 70 min at 4 °C. The final pellet was resuspended in 200-mL PBS [35].

### 4.11. Small Extracellular Vesicles Identification

EVs morphology was determined using a transmission electron microscope (NIKON ECLIPSETI, NIKON, Tokyo, Japan). Simply put, 30-mL EVs samples were dropped onto a copper grid on a 100-mesh paraffin block, dried them with filter paper after 10 min, which was stained with phosphotungstic acid for 15 s and dried at room temperature. The size of EVs was measured by nanoparticle tracking analysis (NTA). Western blot analysis was used to detect EV markers (CD9, TSG101, and CALNEXIN).

### 4.12. Nanoparticle Tracking Analysis (NTA) 

The ZetaView PMX 120 (Particle Metrix, Ammersee, Germany) and its corresponding software (ZetaView 8.04.02) were used to carry out the NTA. Each sample (2 mL) was diluted with PBS 1 time before being loaded into the cell, and then measured at 11 different positions with 2 cycles of readings at each position. After automatically analyzing all these positions and removing any outliers, all data, including the mean, median, mode sizes (diameter), and concentration, were analyzed using this software [36].

### 4.13. Transmission Electron Microscope (TEM)

Briefly, 10 μL of purified EVs were loaded onto copper mesh. The sample was placed at room temperature for 1 min, negatively stained with aqueous phospho-tungstic acid (pH 6.8) for 60 s at room temperature, stained with phosphotungstic acid for 5 min, and subsequently dried under a lamp. Imaging was carried out using a JEM1200EX transmission electron microscope (JEOL, Tokyo, Japan).

### 4.14. Tracking of Small Extracellular Vesicles and Internalization Assay

In the in vitro experiment, after the second step of ultracentrifugation at 4 °C, the EVs were labeled with the green, fluorescent dye PKH67 for 4 min. The complete culture medium was added to stop overstaining. Then, ultracentrifugation was performed at 120,000× *g* at 4 °C for another 70 min. Afterwards, the EVs were incubated with HK-2 cells for 24 h. The cells were then washed with PBS and stained with phalloidin and DAPI [37]. After that, a laser-scanning confocal microscope (LSCM) (NIKON ECLIPSE TI) was used to observe the EVs-HK-2 fusion.

In the in vivo experiment, following the methods of SenRen et al., the tracking of PKH26 labeled HucMSC-EVs in renal tubular epithelial cells was detected [38]. In addition, the visualization and in vivo tracing of PKH26 labeled HucMSC-EVs in the kidney were performed using in vivo imaging systems.

### 4.15. Flow Cytometric Analysis

For the analysis apoptosis, various treatments were performed in different groups. The cells were trypsinized with 0.25% trypsin at 37 °C for 2 min, then centrifuged at 800× *g* for 5 min. The cells were resuspended in PBS, stained with Annexin V-FITC and propidium iodide, and the supernatant was obtained. Lastly, the percentage of cells undergoing apoptosis was analyzed using flow cytometry (FCM) (BD Biosciences, San Jose, CA, USA).

### 4.16. Western Blotting

The cells were lysed with the mixture of RIPA lysis buffer (Beyotime, Shanghai, China) and 1% PMSF (Beyotime, Shanghai, China) for 30 min at 4 °C. A bicinchoninic acid kit (Beyotime, Shanghai, China) was used to quantify protein concentration. An equal amount of protein sample was separated via 12% SDS-PAGE and transferred onto PVDF membranes (Millipore Sigma, St. Louis, CA, USA). Membranes were blocked with 5% skimmed milk at room temperature for 2 h, followed by incubation with the appropriate primary antibodies overnight at 4 °C. The bands were washed 3 times using Tris-buffered saline with Tween, and then incubated with horseradish peroxidase-conjugated secondary antibodies for 1 h at room temperature. Lastly, we used electrochemiluminescence assays to detect protein expression levels.

### 4.17. Statistical Analysis

One-way ANOVA and Student *t*-test were used to analyze group differences. All results are reported as mean ± standard deviation. Further, 2-tailed nonparametric t-tests were used to analyze the differences between two groups, and a value of *p* < 0.05 was considered to be statistically significant.

## Figures and Tables

**Figure 1 ijms-24-08899-f001:**
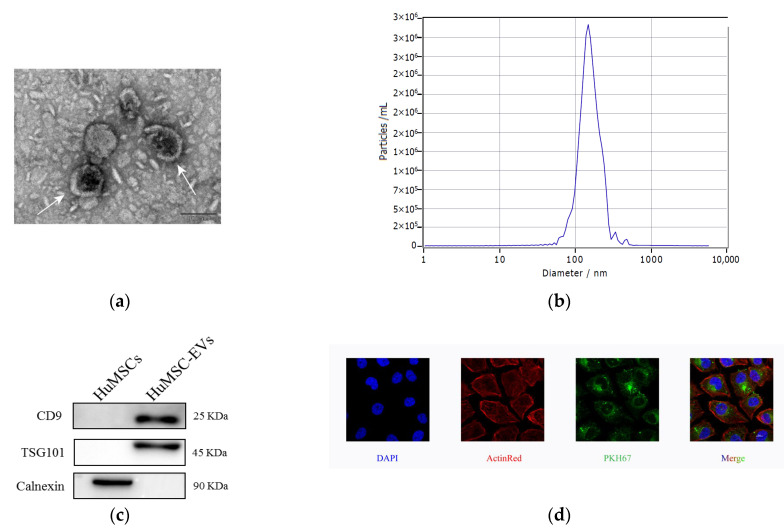
Characterization and internalization of HucMSC-EVs. (**a**) Morphology of HucMSC-EVs was detected by TEM (scale bar = 100 nm). (**b**) HucMSC-EVs size was detected by NTA (mean diameter = 147.5 nm). (**c**) CD9, TSG101 and Calnexin in HucMSCs and HucMSC-EVs were examined by western blot analysis. (**d**) HucMSC-EVs were marked with the green fluorescent dye PKH67 and cytoskeleton and nuclei of HK-2 cells were stained with ActinRed and DAPI. Laser scanning confocal microscope was used to observe. Scale bar = 10 μm (original magnification ×800). DAPI, 4′,6-diamidino-2-phenylindole; HucMSC-EVs, Extracellular vesicles secreted from Human Umbilical Mesenchymal Stem Cells; NTA, nanoparticle tracking analysis; TEM, Transmission electron microscopy.

**Figure 2 ijms-24-08899-f002:**
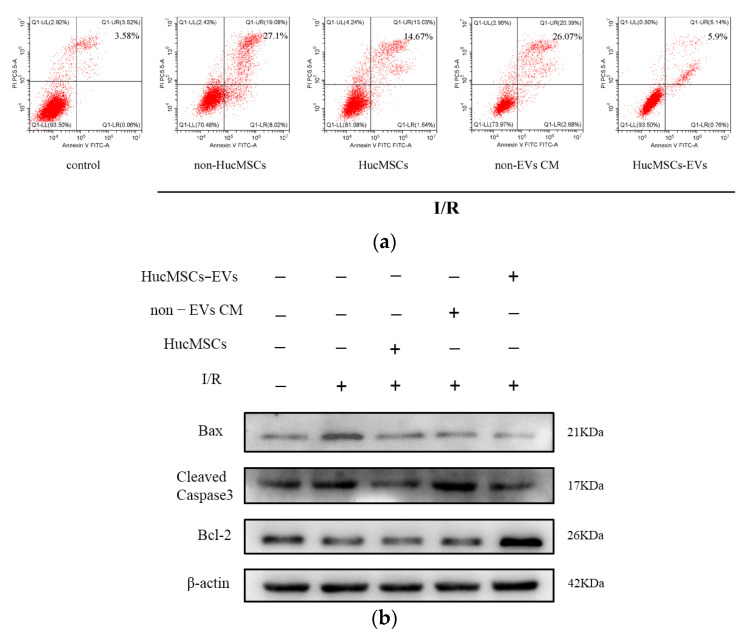
HucMSC−EVs protects HK−2 cells from I/R injury by inhibiting apoptosis. (**a**) HK−2 cells were cocultured with HucMSCs or treated with non−EVs CM or HucMSC−EVs after I/R treatment. Detection of apoptosis rate of HK−2 cells by flow cytometry. (**b**) We co−cultured HK−2 cells with HucMSCs or treated them with non−EVs CM or HucMSC−EVs after I/R treatment. The entire proteins were then extracted and prepared to detect the expression of Bax, cleaved caspase-3, and Bcl-2 by western blotting.

**Figure 3 ijms-24-08899-f003:**
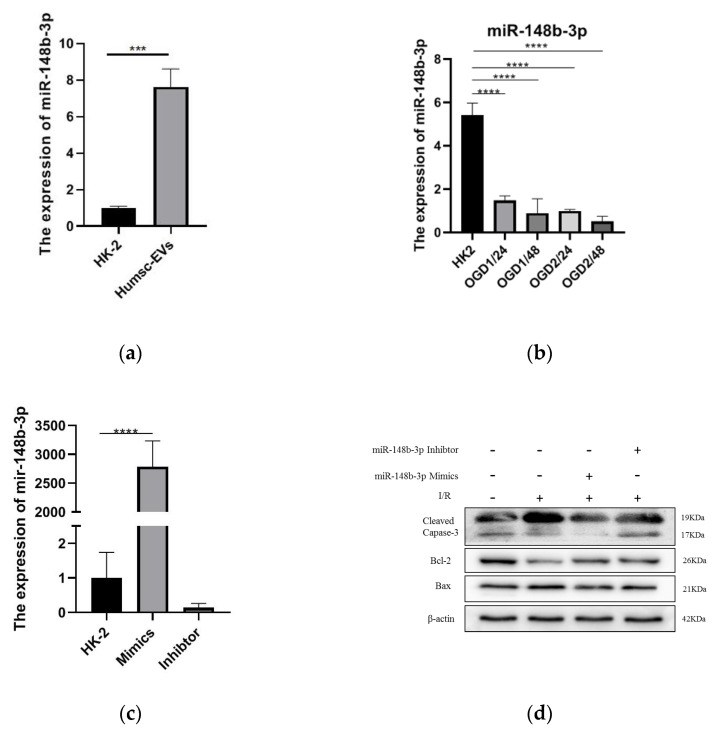
HucMSC−EVs are enriched in miR−148b−3p, which can be delivered into HK−2 cells, thus inhibiting apoptosis induced by I/R injury. (**a**) The total RNA of HK−2 cells and HucMSC−EVs was extracted and reverse transcribed, and the miR−148b−3p level was detceted by qPCR (*** *p* < 0.001 compared with HK−2). (**b**) The expression levels of miR−148b−3p were measured after different treatment times. Ischemia 1 h reperfusion 24 h, ischemia 1 h, 48 h, ischemia 2 h, 48 h. (*** *p* < 0.001 compared with HK−2). (**c**) The qPCR analysis of miR−148b−3p level in HK−2 cells transfected with miR−148b−3p Mimics and miR−148b−3p Inhibitor reagents following ischemia reperfusion treatment (**** *p* < 0.0001 compared with HK-2+NC-M). (**d**) Western blotting was used to detect the effect of miR−148b−3p on apoptosis−related protein expression levels in HK−2 cells treated with overexpression and knockdown. Data are shown as the mean ±SD (*n* = 3). I/R, ischemia/reperfusion; HucMSC−EVs, EVs secreted from human umbilical cord mesenchymal stem cells; NC, negative control; non−EVs CM, non EVs conditioned medium; qPCR, quantitative polymerase chain reaction; SD, standard deviation.

**Figure 4 ijms-24-08899-f004:**
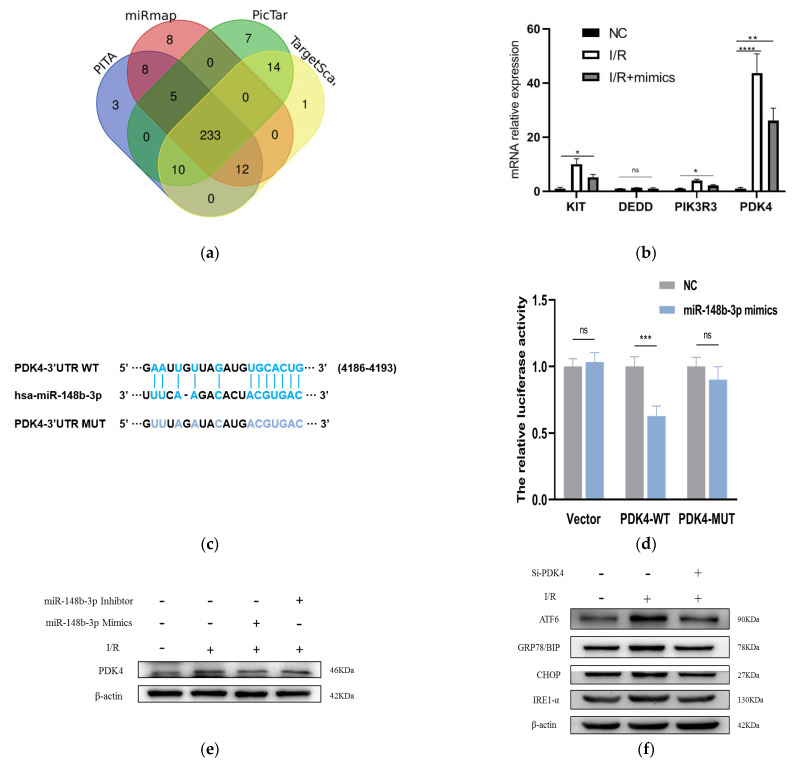
miR-148b-3p allviates the level of *PDK4* and promotes the ER stress and ATF−6 pathways. (**a**) Targeted genes that miR−148B−3p were identified through Pita, PicTar, Target Scan and Miranda. Ten key target genes predicted by these four sites were obtained. (**b**) The levels of *KIT*, *DEDD*, *PIK3R3* and *PDK4* in HK−2 cells cocultured with HucMSCs transfected with miR−148b−3p by PCR analysis (*n* = 3) (* *p* < 0.05, ** *p* < 0.05, *** *p* < 0.001, **** *p* < 0.0001 compared with NC, ns, no significance). (**c**) Schematic diagram depicted the predicted binding site of miR−148b−3p targeting the 3’-UTR of *PDK4*. (**d**) Luciferase reporter assay determined *PDK4* as a bona fide target of miR−148b−3p (*n* = 3). NC, negative control; WT, wide type; MUT, mutant. (**e**) Western blotting was used to detect the effects of NC, miR−148b−3p mimics and miR−148b−3p inhibitors on *PDK4* protein expression in HK−2 cells. (**f**) HK−2 cells and cells transfected with Si−PDK4 were subjected to I/R treatment. The expression levels in ATF−6 pathway of ER stress-related proteins were detected in vitro by western blotting. (**g**) HK−2 cells co−cultured with non−EVs CM, HucMSCs and HucMSC−EVs were treated with I/R injury. The expression levels of ATF−6, GRP78/BIP and CHOP were detected by western blotting. PDC: Pyruvate dehydrogenase complex; *PDK4*: pyruvate dehydrogenase kinase 4; PERK: protein kinase RNA-like ER kinase; IRE−1: inositol−requiring enzyme−1; ATF−6: activating transcription factor 6; CHOP: anti−CCAAT/enhancer−binding protein homologous protein; GRP78/BIP: 78 kDa glucose−regulated protein.

**Figure 5 ijms-24-08899-f005:**
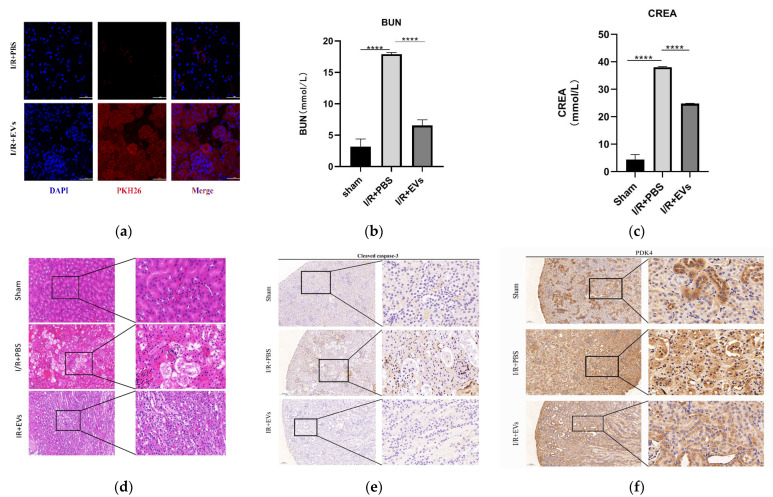
HucMSC-EVs can alleviate renal I/R injury in vivo. (**a**) HucMSC-EVs labeled with PKH26 (red) and DAPI (blue) and injected through tail vein. The mouse of sham group were injected with NS, frozen sections of the kidney were taken 24 h after reperfusion. The sections were stained with DAPI and observed under laser scanning confocal microscope. Scale bar = 50 μm (original magnification: ×600). (**b**,**c**) Blood urea nitrogen and serum creatinine were measured in each group (**** *p* < 0.0001 compared with I/R+PBS group). (**d**) Typical histopathology images of H&E staining in renal sections. Scale bar = 50 μm (original magnification: ×200); scale bar = 20 μm (original magnification: ×400). (**e**,**f**) Representative images of *PDK4* and cleaved caspase-3 immunohistochemical staining in renal tissues (low expression of cleaved caspase-3 and *PDK4* in renal tissue of sham operation group and I/R-treated with PBS group; high expression of *PDK4* and cleaved caspase-3 in I/R-treated with PBS group). Scale bar = 100 μm (original magnification: ×100); scale bar = 20 μm (original magnification: ×400).

**Table 1 ijms-24-08899-t001:** Primer sequences for quantitative real-time polymerase chain reaction.

Genes	Primer Sequence
miR-148b-3p	5′-CGCGTCAGTGCATCACAGAA-3′
U6	5′-ATGGACTATCATATGCTTACCGAT-3′
PDK4	as:5′-CUAUAGACUCAGAAGACAAAG-3′ ss: 5′-CUUUGUCUUCUGAGUCUAUAG-3′
β-actin	Forward:5′-AAACGTGCTGCTGACCGAG-3′ Reverse: 5′-TAGCACAGCCTFFATAGCAAC-3′
human DEDD	Forward:5′-GCCCTACGTCACCCTCAAG-3′ Reverse: 5′-GCCTTGGTTCTGGATCACTG-3′
human KIT	Forward:5′-AAAGGCAGAAGCCACCAACA-3′ Reverse: 5′-AGGAGCGGTCAACAAGGAAA-3′
human PIK3R3	Forward:5′-GGCTTGCCTACCCTGTTCAT-3′ Reverse: 5′-CGGCCTCTCCACTTCACATT-3′
human PDK4	Forward:5′-TGGAGCATTTCTCGCGCTAC-3′ Reverse: 5′-TCCACCAAATCCATCAGGCT-3′

Abbreviations: (PDK4) pyruvate dehydrogenase kinase 4; (DEDD) The death effector domain-containing DNA-binding protein; (KIT) The receptor tyrosine kinase; (PIK3R3) Phosphoinositide-3-kinase regulatory subunit 3.

## Data Availability

Not applicable.

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
