# Peer review of "Human Umbilical Cord Mesenchymal Stem Cells Protect against Renal Ischemia-Reperfusion Injury by Secreting Extracellular Vesicles Loaded with miR-148b-3p That Target Pyruvate Dehydrogenase Kinase 4 to Inhibit Endoplasmic Reticulum Stress at the Reperfusion Stages"

_ijms, 2023, doi:10.3390/ijms24108899_

Round 1
Reviewer 1 Report
Renal ischemia-reperfusion (I/R) injury is a leading cause of acute kidney injury (AKI), 12 with high mortality. In this MS author tested potential of HucMSC extracellular vesicles (HucMSC-EVs) to promote the repair of renal tubular cells. They concluded that HucMSC-EVs protect kidneys from I/R injury during the early I/R stage.
In general, it is a good translational approach to test HucMSC-EVs during IR injury.
Major suggestions:
Sample size is too small n=5/group. For a preclinical study it is highly recommended to use minimum n=3 in triplicates (12) https://www.ncbi.nlm.nih.gov/pmc/articles/PMC3826013/
What is the projected size of these EVs? They may cause microvascular blockage, which may cause localized accumulation. Refer to Figure 5a: shows accumulation, which is not clear at all. I would strongly advice to show a semi quantitive analysis through a bar diagram.
For H&E staining: Captures between 5-10 images within the outer stripe of the outer medulla because it is highly sensitive to hypoxia. So, H&E images with specific labelling is must to confirm if this therapy is working to reduce injury.
There is no evidence if these EVs affect cellular structure of kidney, specially endothelial cells as these cells are impacted during IR injury. There is no mention of tubular injury in IR model as shown in H&E.
Please also discuss about drug loading capacity of these EVs, and how long miR-148b-3p remain stable.
Minor suggestions: I would rather use term shuttle instead of cargo. Refer to line 49 p2
Reviewer 2 Report
* The authors investigated the effects of human umbilical cord mesenchymal stem cells on renal ischemia-reperfusion injury. The topic is interesting but needs language and revision by a specialized company.
* The title should not contain any abbreviations. Please, exchange HucMSCs and PDK4 in your title.
* Line 17: HucMSCs played a protective role and were related to kidney I/R injury. This sentence is not clear.
* Line 20: was predicted online? By which software?
* Include PDK4 in your introduction and the related pathways.
* Figure 2 b, please, complete the legends of the western blot. What are the indicators of the positive and negative signals? please, measure all western blot signals by software like image J quantitatively and represent the results by graphs.
* Figure 5 a, d, e, and f: please, include clear scale bars on the figures. The quantitative measurement of the reaction area by Image J is required.
* Figure 5 e. Is there no caspase-3 reaction? That means total treatment better than negative control itself. Please, revise carefully.
* Line 310: Please, mention all catalog numbers for all used reagents.
* Line 353: please, mention the ethical approval number of your institute.
* Line 407: you should make a histopathologic lesion score for the selected fields.
* Line 419: Please, reveal the results of the analysis.
Reviewer 3 Report
Specific comments
1. Abstract.
The main objective of the study should be added,
The main methods used should be briefely described.
2. Results.
- Line 88. NTA should be defined. (despite being defined in M&M).
- Line 92-92. Authors wrote "Protein markers of EVs (CD9, CALNEXIN, and TSG101) were significantly higher in HucMSC-EVs than in HucMSCs (Figure 1c)."
- According to figure 1C, and since calnexin is considered as a negative marker of small EVs, it should not be higher in HucMSC-EVs than in HucMSCs .
I think the results statement should be revised and corrected.
- Figure 1d should be reloaded with better resolution
- Lines 144-150. By using the figure 3d, It is so difficult to state that there was a difference in apoptosis-related proteins expression after using mimics of inhibitors. A better resolution could fix this.
- Figure 5a should be added in a better resolution.
Discussion.
The results are correctly discussed.
A graphical abstract regarding a suggested model based on the results obtained would be of great interest for the readers.
M&M
Lines 329-338. OGD model shuld have a reference.
Reviewer 4 Report
Comments for the Author (Required):
The study by Shi W et al, titled “HucMSCs protect against renal ischemia-reperfusion injury by secreting extracellular vesicles loaded with miR-148b-3p that target PDK4 to inhibit endoplasmic reticulum stress at the reperfusion stages” aims to understand the beneficial role of HucMSCs-miR-148b-3p on renal ischemia-reperfusion injury. Although this is an interesting article, I have some concerns about this article.
Specific comments
Abstract- line 24. PDK4 and ER expressions induced by I/R injury was significantly inhibited. - It should be PDK4 expressions and ER stress induced…………….
Author needs to quantify all western and Flow cytometry data and quantified data should be represented in bar graph.
Page 3, line 117. Para run like this “In addition, we found that EV-removed CM group was significantly less potent against apoptosis than HucMSCs and HucMSC-EVs”. - Bax and Bcl-2 expression similar when compared to HuMSCs- clarify.
Page 7, line 206. PDK4 protects against MI/R injury- it sounds like PDK4 beneficial against MI/R injury. – double check one more time.
Page 7, line 211. but PDK4-overexpressing reduced the expression of ER stress (Figure 4f). – is it PDK4 overexpression or downregulation (knockdown)?
Page 10, line 325. 4.2 Cell experiments- Is it cell experiments or cell culture experiments?
Page 10, line 336. After 45min were cultured in anoxic medium, fresh oxygenated culture medium- reformulate.
Round 2
Reviewer 1 Report
RE: POINT 1: It would be helpful if you could provide a scientific justification for your response. The first point raised a serious concern regarding your experimental set-up, and provided a reference for your consideration.
Reviewer 2 Report
I have no further comments.
Author Response
Thank you for your helpful comment. We checked the English language and spelling.
Round 3
Reviewer 1 Report
No comments.